# Dynamics of canopy stomatal conductance, transpiration, and evaporation in a temperate deciduous forest, validated by carbonyl sulfide uptake

Richard Wehr[1], Róisín Commane[2], J. William Munger[2], J. Barry McManus[3], David D. Nelson[3], Mark S. Zahniser[3], Scott R. Saleska[1], Steven C. Wofsy[2]

[1]Department of Ecology and Evolutionary Biology, University of Arizona, Tucson, 85721, U.S.A.
[2]School of Engineering and Applied Sciences and Department of Earth and Planetary Sciences, Harvard University, Cambridge, 02138, U.S.A.
[3]Aerodyne Research Inc., Billerica, 01821, U.S.A.

*Correspondence to*: Richard Wehr (rawehr@email.arizona.edu)

**Abstract.** Stomatal conductance influences both photosynthesis and transpiration, thereby coupling the carbon and water cycles and affecting surface-atmosphere energy exchange. The environmental response of stomatal conductance has been measured mainly at the leaf scale, and theoretical canopy models are relied on to upscale stomatal conductance for application in terrestrial ecosystem models and climate prediction. Here we estimate stomatal conductance and associated transpiration in a temperate deciduous forest directly at the canopy scale via two independent approaches: (i) from heat and water vapor exchange, and (ii) from carbonyl sulfide (OCS) uptake. We use the eddy covariance method to measure the net ecosystem-atmosphere exchange of OCS, and we use a flux-gradient approach to separate canopy OCS uptake from soil OCS uptake. We find that the seasonal and diurnal patterns of canopy stomatal conductance obtained by the two approaches agree (to within ±6% diurnally), validating both methods. Canopy stomatal conductance increases linearly with above-canopy light intensity (in contrast to the leaf scale, where stomatal conductance shows declining marginal increases), and otherwise depends only on the diffuse light fraction, the canopy-average leaf-to-air water vapor gradient, and the total leaf area. Based on stomatal conductance, we partition evapotranspiration (ET) and find that evaporation increases from 0% to 40% of ET as the growing season progresses, driven primarily by rising soil temperature and secondarily by rainfall. Counter-intuitively, evaporation peaks at the time of year when the soil is dry and the air is moist. Our method of ET partitioning relies minimally on modeling and avoids concerns about mismatched scales or measurement types because both ET and transpiration are derived from eddy covariance data. Neither of two ecosystem models tested predicts the observed dynamics of evaporation or transpiration, indicating that ET partitioning such as that provided here is needed to further model development and improve our understanding of carbon and water cycling.

## 1 Introduction

Stomata are the adjustable pores through which carbon dioxide ($CO_2$) enters and water vapor exits leaves. Stomata strongly influence both carbon and water cycling, and are a key point of coupling between them (Lin et al., 2015). Stomata also influence surface-atmosphere energy exchange via the latent heat of water vaporization. How readily a leaf's or a canopy's stomata conduct gas is determined by their number and degree of opening, and is quantified by stomatal conductance. At the leaf level, stomatal conductance has been extensively modeled both empirically and theoretically, with both approaches describing it as proportional to net photosynthesis, given fixed $CO_2$ and water vapor concentrations in the air (Collatz et al., 1991; Medlyn et al., 2011). However, scaling leaf-level stomatal conductance up to the canopy is problematic because of the complex heterogeneity of the canopy and its light environment (Bonan et al., 2011). Some empirical evidence suggests that at the canopy scale, stomatal

conductance is not proportional to photosynthesis: as photosynthetically active radiation (PAR) increases, canopy-integrated stomatal conductance (estimated from eddy covariance measurements of ecosystem-atmosphere heat and water vapor exchange) responds linearly while photosynthesis responds nonlinearly (Wehr and Saleska, 2015). A linear response of canopy stomatal conductance to PAR has also been inferred from sap flux measurements (Schäfer, 2011). However, estimates of stomatal conductance based on sap or water vapor flux measurements involve assumptions (e.g. about evaporation, leaf temperature, and plant internal storage) that have contributed substantial uncertainty (Schäfer, 2011; Wehr and Saleska, 2015). Our objective here is to test Wehr and Saleska (2015)'s method for estimating canopy stomatal conductance from the water vapor flux against a new, independent method based on carbonyl sulfide (OCS)—and then to use stomatal conductance to partition evapotranspiration.

OCS is currently a focus of ground- and satellite-based measurements (Kuai et al., 2014; Montzka et al., 2007) and model development (Berry et al., 2013), owing mainly to its potential as a large-scale proxy for gross primary production (GPP) (Blonquist et al., 2011). That potential stems from the fact that OCS is taken up by leaves (Sandoval-Soto et al., 2005), somewhat analogously to $CO_2$. However, while hydrolysis of both OCS and $CO_2$ is catalyzed by the enzyme carbonic anhydrase (CA), the net rate of $CO_2$ hydrolysis depends on downstream reactions involving light while the rate of OCS hydrolysis does not (Berry et al., 2013). Thus the leaf OCS uptake is directly related not to GPP but to CA activity and to the conductance of the diffusive pathway between the air and the chloroplast (Commane et al., 2015). That pathway consists of the leaf boundary layer (i.e. the thin layer of stagnant air on the surface of the leaf), stomata, and mesophyll, with stomata typically being the most influential component (Wehr and Saleska, 2015). At the ecosystem scale, the uptake of OCS by the canopy leaves should be given by:

$$F = gC_n; \quad g = (g_b^{-1} + g_s^{-1} + g_m^{-1} + g_{CA}^{-1})^{-1} , \tag{1}$$

where $F$ is the flux of OCS into the leaves (pmol m$^{-2}$ s$^{-1}$); $C_n$ is the molar mixing ratio of OCS (to air) in the canopy airspace (pmol mol$^{-1}$); $g_b$, $g_s$, and $g_m$ are the canopy-integrated conductances to OCS diffusion through the leaf boundary layer, stomata, and mesophyll, respectively (mol m$^{-2}$ s$^{-1}$); and $g_{CA}$ is the reaction rate coefficient for OCS destruction by CA, expressed as a 'biochemical conductance' (mol m$^{-2}$ s$^{-1}$). Eq. (1) approximates the diffusive pathway as being linear, thereby neglecting possible differences between the parallel pathways associated with individual leaves, as well as the three-dimensionality of diffusion through the mesophyll. This approximation will be justified *a posteriori*.

In addition to being taken up by the canopy, OCS is also taken up or emitted by soils, with the direction and magnitude of the flux depending on soil type, temperature, and moisture (Ogée et al., 2016; Sun et al., 2016; Whelan et al., 2015). The interpretation of large-scale OCS concentrations or fluxes therefore requires understanding of how both canopy and soil processes contribute to the overall ecosystem-atmosphere OCS exchange.

Here we compare canopy stomatal conductance derived from ecosystem-atmosphere heat and water vapor exchange (Wehr and Saleska, 2015) to that derived from canopy OCS uptake via Eq. (1), validating both methods by their agreement. The canopy OCS uptake is measured by using a sub-canopy flux-gradient approach to partition eddy covariance measurements of net ecosystem-atmosphere OCS exchange into canopy and soil components. Mesophyll conductance is estimated using an empirical temperature-dependent function, and boundary layer conductance is estimated using a theoretical model; we are not able to test

these estimates because the gas exchange is not sensitive to them. For the biochemical conductance, we test two simple assumptions: (i) that it is constant and (ii) that it depends on temperature.

Based on the validated canopy stomatal conductance, we partition eddy covariance measurements of total ecosystem-atmosphere water vapor exchange (that is, evapotranspiration, or ET) into transpiration and evaporation, and examine their diurnal and seasonal patterns. Various empirical and theoretical methods have been used to estimate these water fluxes in other ecosystems (Kool et al., 2014), but the present method is advantageous because all fluxes are derived from ecosystem-scale eddy covariance data, minimizing concerns about mismatched scales or measurement types. Finally, we compare the estimated water fluxes to predictions by two ecosystem models.

## 2 Methods

### 2.1 Site description

The data presented here were collected from May to October of 2012 and 2013 at the Harvard Forest Environmental Measurements Site (HF- EMS) in Petersham, Massachusetts, USA. The site is located in a temperate deciduous forest dominated by red oak and red maple (with some hemlock, white pine, and red pine), and has been described in detail previously (Urbanski et al., 2007). Leaf area index (LAI) at the site is steady at about 5 after leaf expansion and before autumnal abscission, although there are inter-annual differences of up to 10% (Wehr and Saleska, 2015).

### 2.2 OCS measurements

The ecosystem-atmosphere exchange of OCS was measured by eddy covariance using a closed-path Quantum Cascade Laser Spectrometer (QCLS) manufactured by Aerodyne Research Inc. and deployed at the site. This QCLS is an improved version of the one described in Commane et al. (2015). The QCLS recorded the OCS concentration in air at 4 Hz with a 1-$\sigma$ precision of 12 parts per trillion (ppt). Measurements were made from May through October in 2012 and 2013.

The QCLS was added to the existing air-conditioned instrument shack, about 10 m from the base of the eddy flux tower, and sampled air in series after a separate QCLS that measured $CO_2$ isotopologue fluxes (Wehr et al., 2013). Air was sampled sequentially from 7 heights on the tower (0.2, 1.0, 7.5, 12.7, 18.3, 24.1, and 29.0 m) through 1∕4″ (outside diameter) Eaton Synflex 1300 tubing with Teflon filters at the inlets to keep out particles. The system's 3-hour duty cycle included 3 eddy covariance measurements (30 min each) separated by profile measurements (7 inlets at 1 min per inlet), as well as 8 soil chamber measurements (5 min per chamber; Sect. 2.3) and 2 QCLS calibrations using gas cylinders (one calibration every 90 min).

The eddy covariance measurements used the 29 m inlet (about 4 m above the top of the canopy), which was on the same boom as the Applied Technologies Inc. sonic anemometer used to measure the wind (this anemometer is part of the site's long-running $CO_2$ system (Urbanski et al., 2007)). The eddy covariance calculations followed the procedure described previously for $CO_2$ isotopologues (Wehr et al., 2013). As the $CO_2$ isotopologue and OCS spectrometers shared the same gas flow system, the co-spectral correction for high-frequency signal attenuation for $CO_2$ was applied also to OCS (the $CO_2$ measurement is much less noisy and so gives a much more precise correction). This approach assumes that OCS and $CO_2$ suffer the same attenuation, which would not be true for a 'sticky' molecule like $H_2O$. Fluxes for periods with a friction velocity of less than 0.17 m s$^{-1}$ were

not used (Urbanski et al., 2007). Storage of OCS in the canopy airspace was accounted for by the profile measurements, which were also used for the flux-gradient approach used to partition canopy uptake from soil uptake (Sect. 2.4).

The gas cylinders contained known amounts of $CO_2$ and negligible OCS; they were used to calibrate the $CO_2$ isotopologue QCLS (Wehr et al., 2013) and to simultaneously calibrate the zero offset of the OCS QCLS. Drift in the zero offset (caused by changes in the QCLS's spectral baseline) was by far the dominant source of error in the raw OCS measurements. To correct for such drift, the zero offset calibrations (one every 90 min) were linearly interpolated in time in post-processing.

## 2.3 Soil chamber $CO_2$ measurements

Of the 8 automated soil chambers sampled, only 4 were used here; these 4 were situated in a 5 x 5 m plot about 50 m to the south of the flux tower. The other 4 were in an adjacent plot that had been trenched to exclude live roots for another study. Each chamber consisted of a polyvinyl chloride (PVC) collar extending 5cm into the ground and covering a surface area of $0.07m^2$. Aboveground, chamber heights were approximately 20 cm. The PVC lid opened pneumatically under the control of a Campbell Scientific CR1000 datalogger. An airtight seal between the lid and collar was achieved by a neoprene gasket on the collar. The chamber design is described in further detail elsewhere (Savage et al., 2014).

The chambers could not be used to measure the OCS efflux from the ground because materials used in the chamber flow system (particularly in the diaphragm pump and the chamber neoprene seal) emitted large amounts of OCS. The system was built for $CO_2$ measurements as part of a separate study and could not practicably be replaced. Thus, the chambers were used instead to determine the $CO_2$ efflux for the flux-gradient method described in Sect. 2.4.

Each chamber was sampled for 5 min once every hour. For the first 1.5 min, the chamber remained open (as it did when not being sampled). For the following 3.5 min, the chamber was closed so that $CO_2$ accumulated in the chamber. A straight line was fit to the rise of the $CO_2$ concentration over time, and the slope of the fit line was used to calculate the $CO_2$ flux into the chamber. Chamber closures for which the coefficient of determination ($r^2$) for the line fit was less than 0.9 were rejected. The $CO_2$ efflux from the soil was calculated as the average of the valid fluxes from the 4 chambers.

## 2.4 Partitioning canopy uptake from soil uptake

A sub-canopy flux-gradient approach was used to partition the net OCS flux into ground and canopy components. This approach assumed that OCS and $CO_2$ were transported to/from the ground by the same turbulent eddies, and that the eddy transport could be treated as gradient-driven, so that the ratio of the gradients of OCS and $CO_2$ just above the ground would be equal to the ratio of their fluxes into the ground:

$$F_{\text{ground}}^{\text{OCS}} = F_{\text{ground}}^{CO_2}(C_{1.0}^{\text{OCS}} - C_{0.2}^{\text{OCS}})/(C_{1.0}^{CO_2} - C_{0.2}^{CO_2}) , \qquad (2)$$

where the $F$ are fluxes into the ground (the $CO_2$ flux is negative) and the $C$ are molar mixing ratios of OCS or $CO_2$ to air, measured at 0.2 m or 1.0 m above the ground, on the flux tower. $F_{\text{ground}}^{\text{OCS}}$ calculated from Eq. (2) was smoothed versus time using a Loess filter, with a time window of about 12 hours used for each local second-order regression. The canopy uptake of OCS was then calculated as the difference between the total ecosystem uptake and the smoothed $F_{\text{ground}}^{\text{OCS}}$. The assumption that the eddy transport near the ground could be treated as gradient-driven is questionable, as turbulent transport in forest canopies is

dominated by large, quasi-periodic eddy motions (Raupach et al., 1996). Nonetheless, the consistency of the calculated soil OCS uptake over the diel cycle and over the growing season (Section 3.1) suggests that the assumption is sufficient for our purposes. Moreover, the calculated soil OCS uptake varied by less than 0.2 pmol m$^{-2}$ s$^{-1}$ between low- and high-turbulence conditions as quantified by the friction velocity (results not shown).

This approach neglects horizontal heterogeneity in the fluxes. In reality, respiration rates vary spatially, and the automated soil chambers used to measure $F_{ground}^{CO_2}$ were located in an area where belowground respiration was on average about 50% higher than in the typical eddy covariance sampling footprint, based on comparison to a multi-year synthesis (Giasson et al., 2013) of over 100,000 soil chamber measurements in that footprint. For this reason, we scaled the measured $CO_2$ efflux from the ground by a

factor of 0.7 to obtain $F_{ground}^{CO_2}$. The flux tower itself, where the gradients were measured, was located about 20 m to the north of the chambers, and the effective footprint of the near-ground concentration measurements is unclear. These spatial mismatches in the sampling footprints introduce additional uncertainty into the mean magnitude of the soil OCS uptake, but should have at most a minor effect on its temporal patterns.

## 2.5 Estimating the diffusive conductances

The leaf boundary layer, stomatal, and mesophyll resistances to $CO_2$ or water vapor transport were estimated (along with leaf temperature) as described for the same forest site in Appendix B of Wehr and Saleska (2015). Each resistance $r$ (s m$^{-1}$) was converted into a conductance $g$ (mol m$^{-2}$ s$^{-1}$) as follows:

$$g = \frac{P}{RT_L}\frac{1}{r},$$   (3)

where $P$ is air pressure (Pa), $R$ is the universal gas constant (8.314 J mol$^{-1}$ K$^{-1}$), and $T_L$ is leaf temperature (K). The resulting conductances for $CO_2$ or water vapor transport were then converted into conductances for OCS transport as follows (Stimler et al., 2010):

$$g_b = g_b^{H_2O}/1.56$$
$$g_s = g_s^{H_2O}/1.94$$   (4)
$$g_m = g_m^{CO_2},$$

where the subscripts $b$, $s$, and $m$ refer to the boundary layer, stomata, and mesophyll, respectively.

For the reader's convenience, we now summarize the equations used to calculate the resistances to $CO_2$ or water vapor transport in Wehr and Saleska (2015). The boundary layer resistance to water vapor transport was calculated as:

$$r_b^{H_2O} = 2\left(\frac{Sc}{Pr}\right)^{\frac{2}{3}}\frac{150}{LAI}\sqrt{\frac{L}{u_h}}\int_0^1 e^{0.5\alpha(1-\zeta)}\phi(\zeta)d\zeta,$$   (5)

where $Sc$ is the Schmidt number for water vapor (0.67), $Pr$ is the Prandtl number for air (0.71), LAI is the leaf area index (i.e. single-sided leaf area per unit ground area, measured as described previously (Wehr and Saleska, 2015)), $L$ is the characteristic

leaf dimension (0.1 m), $u_h$ is the wind speed at the top of the canopy (m s$^{-1}$), $\zeta$ is height as a fraction of canopy top height, $\phi(\zeta)$ is the vertical profile of light absorption normalized such that $\int_0^1 \phi(\zeta)d\zeta = 1$, and $\alpha = 4.39 - 3.97e^{-0.258 \times LAI}$ is the extinction coefficient for the assumed exponential wind profile. The light absorption profile $\phi(\zeta)$ is used here as a proxy for the heat source profile.

Using the leaf boundary layer resistance to heat ($r_b^{heat} = r_b^{H_2O}/0.92$), the leaf temperature was calculated as:

$$T_L = \frac{Hr_b^{heat}}{\rho_a C_p} + T_n \,, \tag{6}$$

where $H$ is the measured eddy flux of sensible heat (W m$^{-2}$), $\rho_a$ is the density of air (kg m$^{-3}$), $C_p$ is the specific heat capacity of air (J kg$^{-1}$ K$^{-1}$), and $T_n$ is the canopy airspace temperature (K). Surface temperatures estimated based on heat flux in this way have been termed "aerodynamic temperatures" to contrast them with radiometric temperatures (Kustas et al., 2007). The canopy airspace temperature $T_n$ was derived from the above-canopy air temperature using a turbulent eddy conductance derived in turn from the corresponding $CO_2$ gradient and NEE. The above-canopy $CO_2$ value was measured at 29 m and the within-canopy $CO_2$

value was the average of measurements at 12.7 m and 18.3 m (canopy top height is about 25 m). The 12.7 m and 18.3 m measurements were generally indistinguishable during the day due to efficient turbulent mixing.

The mesophyll resistance to $CO_2$ transport was calculated as:

$$r_m^{CO_2} = \frac{P}{RT_L}\left(\text{LAI} \times 0.188 \times \exp\left(-\frac{1}{2}\left(\frac{\ln((T_L - 273.15)/28.8)}{0.610}\right)^2\right)\right)^{-1} \,, \tag{7}$$

following the empirical temperature response for *Quercus canariensis* measured by Warren and Dreyer (Warren and Dreyer, 2006), which agrees with the value determined by Manter and Kerrigan (Manter and Kerrigan, 2004) at 25°C for *Quercus rubra* (red oak), which dominates our site (Wehr and Saleska, 2015). There is debate in the literature as to whether mesophyll

conductance varies also in response to light, humidity, and other rapidly changing environmental variables (Flexas et al., 2012; 2008; Tholen et al., 2012). Some studies have argued that mesophyll conductance co-varies with stomatal conductance on a timescale of minutes (Flexas et al., 2008), whereas others have argued that an apparent correlation between the two conductances results from failing to account for diffusion of $CO_2$ from the mitochondria to the chloroplasts (Tholen et al., 2012). Because the stomatal and biochemical conductances limit OCS uptake at our site, changing the mesophyll conductance to be perfectly

correlated with the stomatal conductance (but with a mean value given by Eq. (7)) has a negligible effect on the predicted OCS uptake from Eq. (1). We thus were unable to use OCS fluxes to directly address debates about controls on mesophyll conductance.

The stomatal resistance to water vapor transport was calculated (in the water-flux method) as:

$$r_s^{H_2O} = \frac{1}{\text{LAI}(b_0 e^{b_1 \text{VPG}} e^{b_2 \chi})\text{PAR}} \,, \tag{8}$$

where PAR is incident photosynthetically active radiation ($\mu$E m$^{-2}$ s$^{-1}$), VPG is the water vapor pressure gradient (Pa) between the canopy airspace and the substomatal cavity, and $\chi$ is the clear sky index (i.e. the ratio of observed PAR to the PAR that would have been observed for the same solar zenith angle under clear skies). The coefficients $b_0$, $b_1$, and $b_2$ are empirically determined by fitting Eq. (8) to the following water vapor flux-gradient equation during times when water vapor flux was dominated by transpiration (i.e. evaporation was minimal, typically several days after rain, excluding mornings with possible dew evaporation):

$$r_s^{H_2O} = \frac{VPG}{RT_n E} - r_b^{H_2O} , \tag{9}$$

where $E$ is the measured eddy flux of water vapor (i.e. ET, mol m$^{-2}$ s$^{-1}$). In Eqs. (8) and (9), the vapor pressure gradient is calculated as:

$$VPG = e_{SAT}(T_L) - e_n , \tag{10}$$

where $e_{SAT}(T_L)$ is the saturation vapor pressure (Pa) at temperature $T_L$ and $e_n$ is the vapor pressure (Pa) in the canopy airspace. With $b_0 = 1.442 \times 10^{-4}$, $b_1 = -4.035 \times 10^{-4}$, and $b_2 = -0.8473$, Eq. (8) was able to explain 92% of the variability ($r^2 = 0.92$) in hourly evapotranspiration over three entire growing seasons, during periods when evaporation was minimal (Wehr and Saleska, 2015). It was necessary, however, to use wind-direction-dependent LAI because of ecosystem heterogeneity surrounding the HF-EMS site, as described in Wehr and Saleska (2015). The clear sky index is used as a convenient proxy for diffuse light fraction in Eq. (8); when available, measurements of the diffuse light fraction can be used instead.

Eqs. (6,9-10) can be combined to obtain the Penman-Monteith (PM) equation (Monteith et al., 1965), which is commonly used to calculate stomatal conductance. In principle, the two approaches are equivalent. However, the PM equation expresses $H$ as the residual of the other terms in the energy budget (i.e. net radiation $R_n$, latent heat flux $\lambda E$, conductance to ground $G$, and energy storage $S$) because measurements of $H$ were not available when the PM equation was derived. As $H$ is measured at all eddy flux sites, using the PM equation to calculate stomatal conductance from eddy flux measurements of $E$ is unnecessarily circuitous. Moreover, it yields a different result in practice because the measured energy budget is incomplete at most eddy flux sites, including ours, with the radiative input exceeding the turbulent output by about 20%. This energy balance closure problem has been repeatedly investigated but the culprit remains unclear and might vary between sites (Wilson et al., 2002; Foken, 2008; Lindroth et al., 2010). If the closure problem were known to be due to low bias in the eddy covariance method, as suggested by Foken (2008), then it would be appropriate to adjust $H$ and $E$ to satisfy energy balance while maintaining the measured Bowen ratio ($H/\lambda E$), either by modifying the PM equation as in Wohlfahrt et al. (2009) or by multiplying $H$ and $E$ by ($R_n - G - S$)/($H + \lambda E$) in Eqs. (6,9); however, recent evidence suggests that the problem is due to energy storage in the soil and biomass instead (Lindroth et al., 2010). At our site, the original PM equation underestimates stomatal conductance by 20% compared to Eqs. (6,9-10).

Eqs. (8-9) constitute the water-flux method for estimating stomatal conductance. To obtain stomatal conductance from our OCS measurements, we rearranged Eq. (1) to solve for $g_s$ and inserted the measured canopy OCS uptake $F$ along with the boundary layer, mesophyll, and biochemical conductance estimates described in this section and in Section 2.6. As mentioned above (and

see Section 3.2), the stomatal and biochemical conductances were limiting to OCS uptake at our site, and so the calculated stomatal conductance was not sensitive to the boundary layer and mesophyll conductances used.

At night, the leaf-air water vapor gradient was too small to drive measurable transpiration, and so Eqs. (8) and (9) could be used to determine nighttime $r_s$. Instead, we set nighttime $r_s$ so as to obtain agreement between predicted and observed nighttime canopy OCS uptake (Sect. 3.2). Nighttime stomatal conductance turned out to be small enough that the canopy OCS uptake was completely insensitive to all other parameters within their plausible bounds.

The OCS- and water-flux methods for estimating stomatal conductance are illustrated as a flow chart in Fig. 1.

$$r_m^{CO_2} = \frac{P}{RT_L}\left(\text{LAI}\times 0.188\times\exp\left(-\frac{1}{2}\left(\frac{\ln((T_L-273.15)/28.8)}{0.610}\right)^2\right)\right)^{-1}$$

## 2.6 Modeling the carbonic anhydrase activity

The one term in Eq. (1) that is not constrained by measurements, empirical models, or established theory is the biochemical conductance associated with carbonic anhydrase activity, $g_{CA}$. Apparent CA activity depends on the amount of CA enzyme and on where it is located relative to the intercellular air spaces, but little is presently known about either of those things (Berry et al., 2013). We therefore tested two simple assumptions for $g_{CA}$ that allow us to solve Eq. (1). First, we assumed CA activity to be constant at the leaf scale, with the value 0.055 mol m$^{-2}$ s$^{-1}$ chosen to give closest agreement with the observations; thus:

$$g_{CA} = 0.055\times\text{LAI}, \tag{11}$$

Second, we assumed $g_{CA}$ to follow an Arrhenius function of leaf temperature:

$$g_{CA} = 0.8(0.055\times\text{LAI})e^{\frac{E_0}{R}\left(\frac{1}{T_{ref}}-\frac{1}{T_L}\right)}, \tag{12}$$

where $E_0$ (40 J mol$^{-1}$) is the activation energy and the additional factor of 0.8 was chosen to maintain agreement with the observations at noon given the arbitrary reference temperature $T_{ref}$ = 20°C. The magnitude of $E_0$ controls the magnitude of the temperature dependence, but the choice of $E_0$ does not impact our findings; the value 40 J mol$^{-1}$ was chosen merely to be large enough to distinguish the predictions based on Eqs. (11) and (12).

## 2.7 Partitioning evapotranspiration

We calculated transpiration for the whole data record by inverting Eq. (9) to solve for the water flux $E$ and inserting the stomatal resistance calculated from Eq. (8). Evaporation was then calculated as the difference between transpiration and total measured water flux (that is, ET). This method of partitioning ET is advantageous because both ET and transpiration are derived from eddy

covariance data, minimizing concerns about mismatched scales or measurement types. Such concerns arise when using sap flux sensors or gas exchange chambers to estimate transpiration, for example (Kool et al., 2014).

## 3 Results and Discussion

### 3.1 Soil-atmosphere exchange of OCS

The soil was a consistent sink of OCS at this site (Figs. 2 and 3). OCS uptake by soil was highly stable, with a mean of 2.2 pmol $m^{-2}$ $s^{-1}$ and no discernable diel or seasonal patterns (for May through October). That value is roughly in the middle of the range reported for temperate forest soils (-8 to +1.45 pmol $m^{-2}$ $s^{-1}$) (Whelan et al., 2015). Laboratory incubations of soy field soil have shown OCS uptake to depend strongly on soil temperature and water content (Whelan et al., 2015), but we see no indication of those dependencies in the field at our site. It may be that the dependencies are difficult to detect because the ranges of soil

temperature (~9-18°C) and volumetric water content (~22-37%) are too small. It may also be that the dependencies on temperature and water offset one another: OCS *production* increased with temperature in the incubations (Whelan et al., 2015) and might therefore be expected to be highest midseason, while OCS *consumption* peaked at roughly 20% water content in the incubations (Whelan et al., 2015) and might therefore also be expected to be highest midseason at our site.

### 3.2 Canopy stomatal conductance and OCS uptake

Nighttime uptake of OCS by the canopy averaged 2.4 pmol $m^{-2}$ $s^{-1}$, consistent with a nighttime stomatal conductance to OCS transport of 0.0016 mol $m^{-2}$ $s^{-1}$ on a leaf area basis (about 1/40 of the mean midday value). We used this value to generate the nighttime points on the predicted uptake curve in Fig. 2. The corresponding conductance to water vapor transport is 0.0031 mol $m^{-2}$ $s^{-1}$. Nighttime stomatal conductance to water vapor transport in individual red oak and red maple leaves at this site was previously reported to be no more than 0.05 mol $m^{-2}$ $s^{-1}$ (Daley and Phillips, 2006), in agreement with our result. At another site,

measured values ranged from 0 to 0.039 mol $m^{-2}$ $s^{-1}$ with an average of 0.011 mol $m^{-2}$ $s^{-1}$ (Barbour et al., 2005), which is about 4 times larger than found here using OCS. Previous estimates of nighttime *ecosystem* OCS uptake at this site averaged about 4 pmol $m^{-2}$ $s^{-1}$ between April and November (Commane et al., 2015), similar to the 4.6 pmol $m^{-2}$ $s^{-1}$ (= 2.2 soil + 2.4 canopy) found here.

During the day, Eq. (1) successfully predicted the diurnal and seasonal patterns of canopy OCS uptake when CA activity was assumed constant at 0.055 mol $m^{-2}$ $s^{-1}$ on a leaf area basis (Figs. 2 and 3, dotted lines), except at dawn (see below). Apart from dawn, the agreement with observations was within ±3% for the diurnal pattern and ±10% for the seasonal pattern (1 standard deviation). Assuming CA activity to be a function of temperature did not improve the agreement (Figs. 2 and 3, dashed lines). Rearranging Eq. (1) to solve for $g_s$ using the measured canopy OCS uptake gave a diurnal cycle of stomatal conductance that

agreed with that obtained from Eq. (8) to within ±6% (1 standard deviation), again apart from dawn (Fig. 4).

At dawn, no value of CA activity or mesophyll conductance or boundary layer conductance was capable of bringing the predicted canopy uptake into agreement with the observations, because the stomatal conductance was strongly limiting at that time of day (Fig. 4). To obtain agreement, the stomatal conductance would have had to be 5 times larger than estimated from the

measured heat and water fluxes, which is implausible. We therefore suspect that the apparent surge in canopy OCS uptake at dawn is a measurement artifact resulting from a failure of the vertical profile measurements on the flux tower to adequately capture changes in OCS storage within the canopy and subcanopy airspaces in the eddy covariance footprint (which is tens to

hundreds of meters upwind of the tower). Such a failure could occur because of differences in storage between the eddy covariance footprint and the profile footprint, and/or because of hidden OCS storage in unsampled airspaces, such as in the leaf litter layer or within tree crowns. Specifically, we speculate that our vertical profile measurements on the tower do not adequately capture the depletion of the OCS concentration in the canopy and subcanopy airspaces overnight and—more importantly here—the replenishing of that concentration with increased turbulence at dawn. Thus eddy flux that is in reality serving to replenish the OCS concentration is misinterpreted as being taken up by the leaves. We have sometimes observed a similar phenomenon in $CO_2$, but with reversed sign: $CO_2$ accumulates in the canopy and subcanopy airspaces overnight and is flushed out at dawn, so that if the changes in canopy and subcanopy $CO_2$ concentration are not adequately measured, there appears to be a dawn pulse in respiration and in the net ecosystem-atmosphere $CO_2$ exchange.

The total conductance $g$, which controls the OCS uptake, is limited by $g_s$ early and late in the day but co-limited by $g_s$ and $g_{CA}$ through the middle of the day, so that $g$ has a much flatter diurnal peak than $g_s$ (Fig. 4; note the log scale). Eq. (1) must therefore include the correct diurnal patterns and magnitudes of both stomatal conductance and CA activity in order to reproduce the observed diurnal pattern of canopy OCS uptake. Notwithstanding the dawn artifact, then, the close agreement between the temporal patterns of observed and predicted stomatal conductance in Fig. 4 (or total conductance in Fig. 4 or OCS uptake in Fig. 2) constitutes a strong validation of the simple empirical description of canopy stomatal conductance in Eq. (8), which should provide a reliable means to partition ET at this mesic forest site.

Eq. (8) was distilled from a more general equation allowing for direct temperature and soil moisture effects on $g_s$, and for a saturating response of $g_s$ to PAR, but those effects were found to be negligible (Wehr and Saleska, 2015). The linear response of (canopy-scale) $g_s$ to PAR in Eq. (8) contrasts with the saturating response generally observed at the leaf scale, and implies that leaf-level stomatal models like the Ball-Berry model (Collatz et al., 1991) or models based on optimality theory (Medlyn et al., 2011) should not be applied directly at the canopy scale (as in a 'big leaf' canopy model; De Pury and Farquhar, 1997), at least at this site. Note that even though canopy $g_s$ responds linearly to PAR, it will appear to show a saturating response when the vapor pressure gradient is not controlled for, as PAR and the vapor gradient tend to be positively correlated (because clouds and moisture go together) while $g_s$ and the vapor gradient are negatively correlated (because stomata close to limit plant water loss). The insensitivity of $g_s$ to soil moisture is somewhat surprising, but as Harvard Forest is a mesic site, it might be that soil water does not become limiting to the large trees even in a dry year such as 2012. The insensitivity to temperature, on the other hand, is not surprising: it has been argued (Lloyd and Farquhar, 2008) that photosynthesis declines at high temperature almost entirely due to the indirect effect of temperature on stomatal conductance via the leaf-to-air water vapor pressure gradient, and Eq. (8) is consistent with that view.

The boundary layer conductance $g_b$ is likely overestimated at night in Fig. 4 because the value from Eq. (5) is weighted vertically according to the profile of light absorption calculated from the leaf area density profile and the solar zenith angle—a method appropriate to sunlight-driven heat and $CO_2$ fluxes during the day but not to any flux at night. Because the stomatal conductance is negligible at night and therefore strongly limiting, this overestimation of $g_b$ is inconsequential here.

While treating CA activity as constant on a leaf area basis was sufficient to reproduce the observed canopy OCS uptake at this site, seasonal variations in CA activity at the ±15% level could also be consistent with our observations, and might be important for some studies.

We did not observe during 2012-2013 any repeat of the anomalous OCS emission previously reported at this site for July, 2011 (Commane et al., 2015), despite similar environmental conditions. The cause of that apparent emission remains unknown.

### 3.3 Gross Primary Productivity and Leaf Relative Uptake

The success of Eq. (1) in predicting canopy OCS uptake supports the theoretical expectation that canopy OCS uptake is dependent on the atmospheric OCS mixing ratio, the diffusive conductance, and CA activity—and is therefore not directly related to GPP, which depends also on the enzyme Rubisco and on light. There is nonetheless widespread interest in using OCS uptake as a proxy for GPP. So for interested readers, we show how GPP and OCS uptake compare over the diel cycle and over the growing season in Figs. 5 and 6, using both isotopic and standard partitioning methods for estimating GPP as described in

Wehr et al. (2016). We include a common metric for comparison, the leaf relative uptake (LRU): LRU is defined as the ratio of the canopy OCS uptake to GPP, divided by the ratio of the atmospheric OCS concentration to the atmospheric $CO_2$ concentration. Figs. 5 and 6 show that, as expected, LRU is inversely correlated with PAR, as PAR can directly limit uptake of $CO_2$ but not of OCS.

### 3.4 Transpiration and evaporation

On average over the growing season, evaporation accounted for 16% of ET; however, the proportion increased strongly through the growing season, from 0% in May to 40% in October (Fig. 7). This trend was mostly due to increasing evaporation rather than decreasing transpiration, except in October, when leaves began to fall. Perhaps counter to intuition, evaporation peaked at the time of year when the soil was dry (low volumetric soil moisture) and the air was moist (low vapor pressure deficit). There are two reasons. First, rainwater that collects on the leaf litter layer (Wilson et al., 2000), exposed soil surfaces, and canopy surfaces

contributes to evaporation but is not affected by transpiration, which draws down the volumetric soil moisture in summer. Second, evaporation from the ground is driven by the water vapor pressure gradient between the ground and the atmosphere, which depends not only on the available liquid water and the relative humidity, but also on the temperatures of the ground and atmosphere. For a given amount of liquid water or relative humidity, the vapor pressure increases with temperature.

In light of the above, we can identify different controls on the ratio of evaporation to ET in different seasons (color-coded in the lower panels of Fig. 7): in spring, despite plenty of rain and soil moisture, evaporation was suppressed by the negative soil-to-air vapor pressure gradient that resulted from the lagged rise of soil temperature relative to the atmosphere (red); in summer, evaporation was mostly limited by rainfall (blue); and in autumn, evaporation remained steady but the ratio of evaporation to ET increased because transpiration declined with LAI (yellow). There was also a period in September when the ratio of evaporation

to ET remained constant despite declining rainfall, due again to the lagged rise of soil temperature (red). The overall increasing trend in evaporation through the season was driven primarily by soil temperature rather than by rain, soil moisture, or atmospheric vapor pressure deficit (VPD). A similar pattern in evaporation was observed from the ground surface of another temperate deciduous forest via sub-canopy eddy covariance measurements (Wilson et al., 2000), but soil temperature was not discussed as a driver of that pattern.

On average, evaporation and the ratio of evaporation to ET were highest immediately following rain and then declined (Fig. 8). ET, on the other hand, increased with time since rain, likely because of drier air, increased sunlight, and associated warmer surface temperatures.

Although our transpiration and evaporation estimates are not direct measurements, only three sets of assumptions are common to both of the methods by which we derived those estimates from our measurements: (1) the assumptions involved in the eddy covariance method, which seem to fail at dawn as discussed in Sect. 3.2; (2) the boundary layer resistance model described in Sect. 2.5, to which our estimates are insensitive (within plausible bounds); and (3) the application of stoma-scale gas transport equations at the canopy scale, which neglects biological and environmental heterogeneity within the canopy but is a sufficiently good approximation to allow the observed temporal patterns of canopy OCS uptake to be precisely predicted from the observed heat and water vapor exchange, as shown in Sect. 3.1. Given the close agreement between the two otherwise independent methods (Sect. 3.2), our estimates of transpiration and evaporation should be accurate.

The stomatal and evaporative dynamics found here can therefore be used to evaluate and improve the representation of carbon and water cycling in biosphere models. For example, Fig. 7 compares our empirical water flux estimates to predictions by the Ecosystem Demography 2 (ED2) (Medvigy et al., 2009) and Simple Biosphere 3 (SiB3) (Baker et al., 2008) models for the same site and exactly the same set of hourly measurement intervals. (The ED2 and SiB3 parameters were optimized for the site using standard data that included NEE and ET but not our partitioned water fluxes or OCS measurements.) ED2 reproduces the seasonal variation of measured ET well, but for the wrong reasons: it underestimates transpiration by a factor of 2 and greatly exaggerates evaporation, particularly in spring. Moreover, ED2 predicts transpiration to fall off too rapidly in spring and autumn. SiB3 underestimates transpiration by just 20% and evaporation by just 25%, on average, but does not predict the early summer peak in transpiration or its persistence in autumn. Neither model predicts the observed increase in evaporation through the growing season, the magnitude of which is equal to 25% of summertime ET. We do not troubleshoot the biosphere models here; suffice it to say that the stomatal and evaporative dynamics in this ecosystem are not accurately represented, and that our findings and methods provide new means to diagnose and improve those aspects of the models.

**4 Conclusion**

Partitioning the net ecosystem-atmosphere exchange of OCS into canopy and soil components revealed that the soil was a consistent sink of OCS at this mesic, temperate deciduous forest site, while the canopy took up OCS according to a simple prediction based on the OCS concentration in the air, the estimated diffusive conductance from the air to the chloroplast, and the assumption of constant carbonic anhydrase activity on a leaf area basis.

Our findings validate: (i) a simple empirical function for canopy stomatal conductance at this site, which depends linearly on PAR (in contrast to the leaf scale) and otherwise depends only on the leaf-to-air water vapor gradient, the diffuse light fraction, and the leaf area index; (ii) the approach used to derive that function from eddy covariance measurements of heat and water vapor exchange (Wehr and Saleska, 2015); and (iii) the assumption that CA activity can be treated as constant over time in this ecosystem. Our findings also suggest that OCS flux measurements could serve as a general tool for probing stomatal conductance in other ecosystems, and possibly in leaf-level and laboratory studies.

Using the empirical function for stomatal conductance, we partitioned evapotranspiration and found that evaporation peaked in late summer when the soil was dry and VPD was low, mostly due to the strong influence of soil temperature on the evaporation rate. From May to October, evaporation increased from 0% to 40% of evapotranspiration, with a mean of 16%. Neither of the

two ecosystem models that we tested predicted the observed increase of evaporation through the growing season, and neither accurately predicted the seasonal dynamics of transpiration. Through empirical quantification of canopy stomatal conductance, transpiration, and evaporation, the methods described here enable more stringent evaluations of model performance, and improve our ability to predict carbon and water cycling.

## 5 Author contributions

J. B. McManus, D. D. Nelson, M. S. Zahniser, R. Commane, and S. C. Wofsy developed the OCS spectrometer. R. Commane deployed and managed the spectrometer at the forest with the assistance of R. Wehr and J. W. Munger. R. Wehr wrote the data processing and analysis algorithms and analysed the results with input from R. Commane, J. W. Munger, and S. R. Saleska. R. Wehr prepared the manuscript with contributions from all co-authors.

## 10 Competing interests

The authors declare that they have no conflict of interest.

## Acknowledgments

This research was supported by the US Department Of Energy (DOE)'s Office of Science's Terrestrial Ecosystem Science program (award DE-SC0006741) and the Agnese Nelms Haury Program in Environment and Social Justice at the University of 15 Arizona. Design of the OCS spectrometer was supported by the DOE's Small Business Innovation Research program (award DE-SC0001801). Operation of the HFEMS flux tower is supported by DOE via the Ameriflux network, and the tower is a component of the Harvard Forest Long-Term Environmental Research (LTER) site supported by the US National Science Foundation (NSF). Soil chamber $CO_2$ flux data were provided by Kathleen Savage at the Woods Hole Research Center and Eric Davidson at the University of Maryland Center for Environmental Science, also supported by award DE-SC0006741. Soil 20 temperature and soil moisture data were provided by E. Nicoll at the Harvard Forest, supported by the NSF LTER program. ED2 model output was provided by Daniel Scott and Paul Moorcroft at Harvard University, also supported by award DE-SC0006741. SiB3 model output was provided by Ian Baker at Colorado State University, supported by the US National Oceanic and Atmospheric Administration's Atmospheric Chemistry, Carbon Cycle and Climate program, the US National Aeronautics and Space Administration's Carbon Cycle Science program, and the NSF's Science and Technology Center for Multiscale Modeling 25 of Atmospheric Processes, managed by Colorado State University.

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

**Figures and captions**

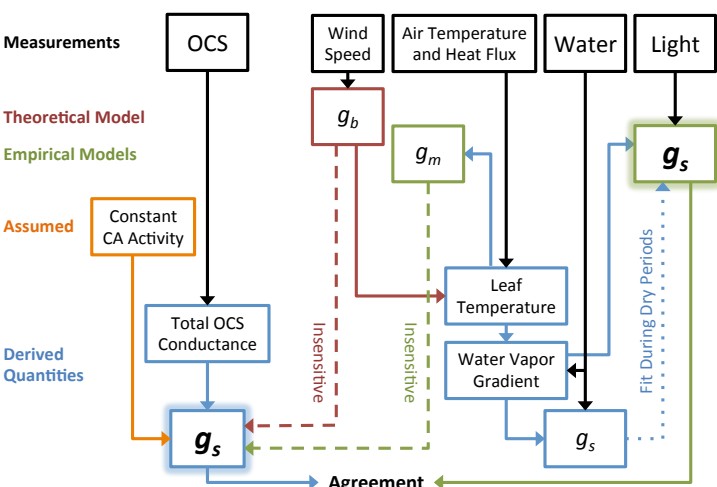

**Figure 1. Flowchart of the OCS- and water-flux methods for estimating stomatal conductance. Note that the assumed CA activity and**
25  **the empirical models for $g_b$, $g_m$, and $g_s$ also depend on leaf area index (not shown).**

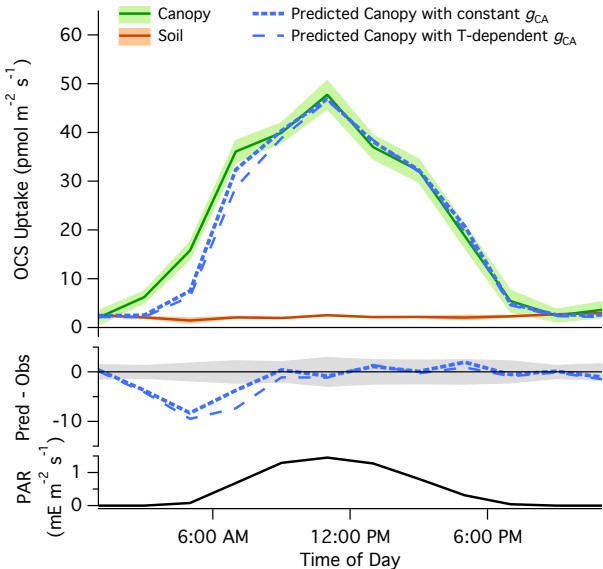

**Figure 2: Composite diel cycles (including May through October of 2012 and 2013) of soil uptake, canopy uptake, and predicted canopy uptake from Eq. (1), along with the differences between predicted and observed canopy uptake, and photosynthetically active radiation (PAR) for reference. Lines connect the mean values of each 2-hour bin. Pale bands depict standard errors in the means as estimated from the variability within each bin.**

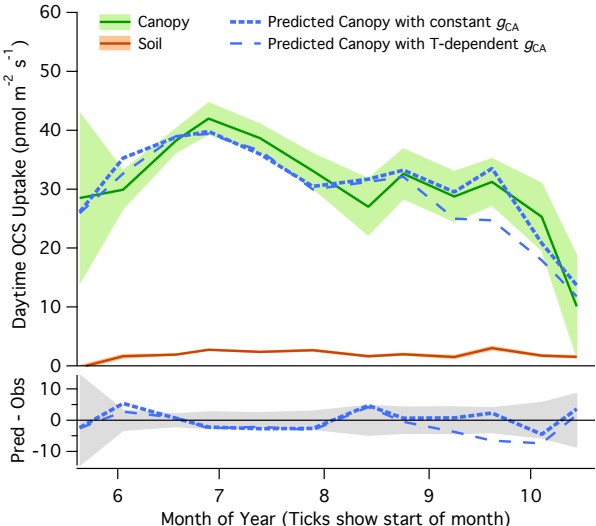

**Figure 3: Composite seasonal cycles (including the years 2012 and 2013) of daytime soil uptake, canopy uptake, and predicted canopy uptake from Eq. (1), along with the differences between predicted and observed canopy uptake. Lines connect the mean values of each 2-week bin. Pale bands depict standard errors in the means as estimated from the variability within each bin.**

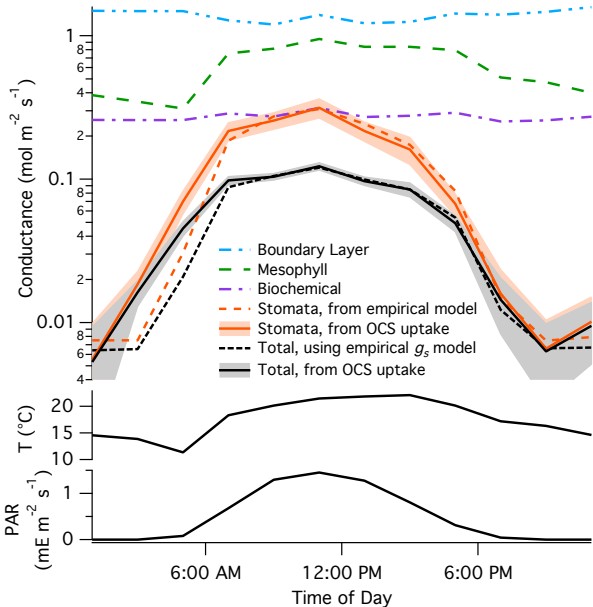

**Figure 4: Composite diel cycles (including May through October of 2012 and 2013) of the expected leaf boundary layer, stomatal, mesophyll, biochemical, and total conductances to OCS transport and chemical destruction, along with the stomatal and total conductances derived from OCS uptake. Lower panels show air temperature and PAR for context. Lines connect the mean values of each 2-hour bin. Pale bands depict standard errors in the means as estimated from the variability within each bin. The boundary layer conductance is likely overestimated at night (see text).**

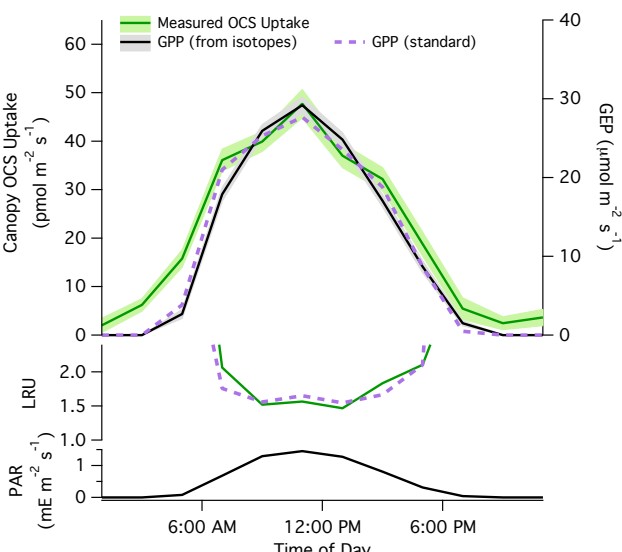

**Figure 5. Composite diel cycles (including May through October of 2012 and 2013) of canopy OCS uptake and GPP, along with leaf relative uptake (LRU) and photosynthetically active radiation (PAR). GPP from the isotopic and standard partitioning methods described in Wehr et al. (2016) are shown. Lines connect the mean values of each 2-hour bin. Pale bands depict standard errors in the means as estimated from the variability within each bin.**

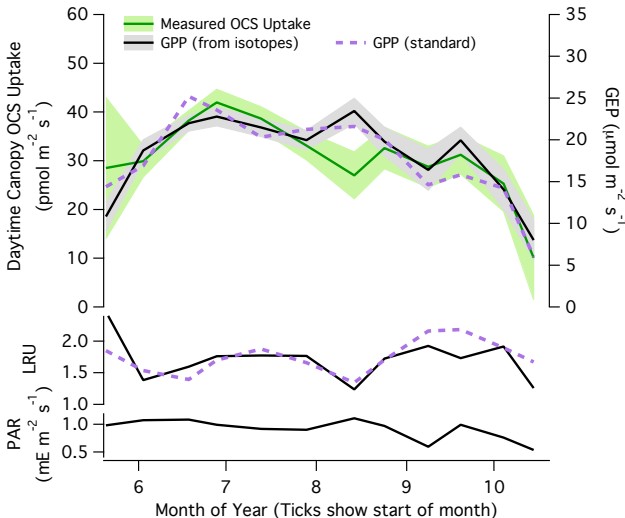

**Figure 6. Composite seasonal cycles (including the years 2012 and 2013) of canopy OCS uptake and GPP, along with leaf relative uptake (LRU) and photosynthetically active radiation (PAR). GPP from the isotopic and standard partitioning methods described in Wehr et al. (2016) are shown. Lines connect the mean values of each 2-week bin. Pale bands depict standard errors in the means as estimated from the variability within each bin.**

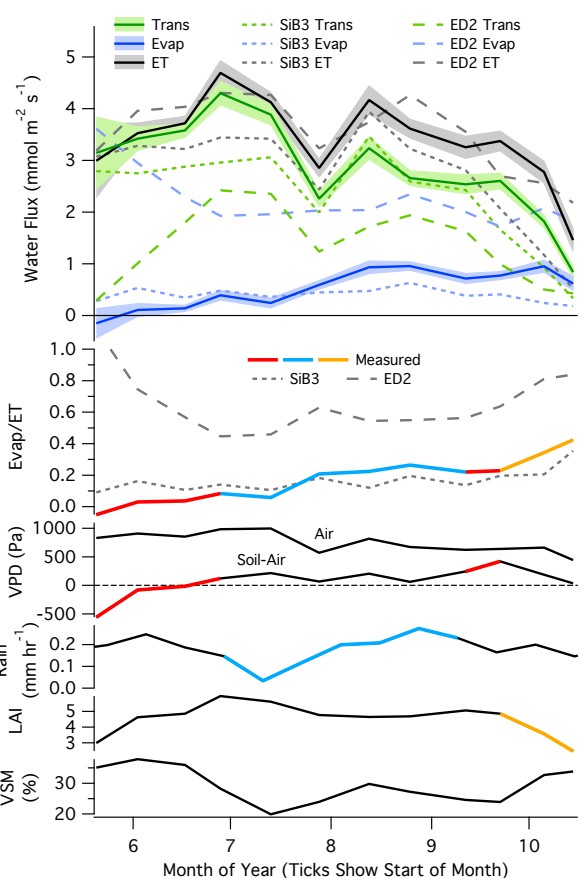

**Figure 7: Composite seasonal cycles (including the years 2012 and 2013) of daytime evapotranspiration (ET), transpiration, and evaporation, as estimated here and as predicted by the ED2 and SiB3 models, along with: the ratio of evaporation (Evap) to ET, the atmospheric vapor pressure deficit (VPD), the soil-atmosphere vapor pressure difference, rainfall, the leaf area index (LAI), and the volumetric soil moisture content between 0 and 30 cm depth (VSM). Lines connect the mean values of each 2-week bin. Pale bands depict standard errors in the means as estimated from the variability within each bin. The soil-air vapor pressure difference is based on the assumption of saturated air at the soil surface and therefore reflects the maximum possible difference. Color-coding in the lower panels highlights which environmental control is dominating the behavior of the Evap/ET ratio. Model output was filtered to include only times when measurements were made.**

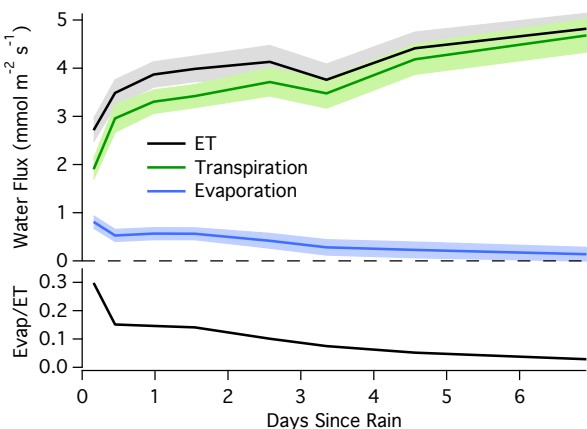

**Figure 8. Daytime evapotranspiration, transpiration, evaporation, and the ratio of evaporation to evapotranspiration versus time since the cessation of rain.**