# Peer review of "Dynamics of canopy stomatal conductance, transpiration, and evaporation in a temperate deciduous forest, validated by carbonyl sulfide uptake"

_Biogeosciences, 2016_

## Referee Comment (RC1) · Anonymous Referee #1 · 12 Sep 2016

This is one of, if not the, first study to use carbonyl sulfide (COS) to investigate canopy-scale stomatal conductance and transpiration (and by difference from evapotranspiration, evaporation). The authors are modelling the various resistances to canopy COS exchange, which they infer from ecosystem-scale COS flux measurements and estimates of the soil exchange, to finally back out the stomatal resistance/conductance. These estimates are compared against estimates derived from a combination of sensible and latent heat flux measurements. The two approaches agree reasonably well, except for around dawn, when the authors suspect measurement artefacts. Two, random, land surface models do a poor job in simulating the magnitude, partitioning and

seasonal evolution of the water fluxes, highlighting the value of the presented data for improving models. This is an excellent paper, well written and innovative. The comments below serve to further improve the manuscript.

Major comments:

(1) In contrast to the authors I believe that their approach does not rely "minimally on modelling" (p. 1, l. 24), but rather that they use a complex model to simulate tremendously complex processes. In particular the vertical heterogeneity of the microclimate and associated patterns in exchange processes constitute a real challenge for upscaling. I found the description of the modelling approaches sometimes hard to follow, in particular with regarding to keeping track what the different inputs are and by which approach these are used and what the assumptions are. In order to make this easier for the reader to follow I am suggesting a sort of a flow chart or similar that shows the inputs for the two approaches, lists the main equations and outputs and so makes it easier to follow what the differences between the approaches are.

(2) I am surprised to see that what I would consider the standard approach for assessing the surface conductance to water vapour, the Penman-Monteith combination equation, was not used. This approach is used a lot in the flux measurement community and it would be nice to see how it compares to the other approaches. In doing so, and the same applies to the approach based on sensible and latent heat fluxes, the authors will need to deal with the energy imbalance (if it exists at this site, which I though presume).

(3) In my view any paper should have a spelled-out statement of objectives and/or hypothesis and I am asking the authors to modify the introduction paragraph accordingly.

Detailed comments:

(1) p. 1, l. 32: and degree of opening

(2) p. 2, l. 13: $CO_2$ assimilation depends both on the light AND dark reactions

(3) p. 2, l. 16: stomata are generally the most influential component

(4) p. 3, l. 31: that will work if both COS and CO2 suffer the same attenuation; for CO2 and H2O we know that this does not work very well, H2O being more strongly attenuated in the inlet tubes

(5) p. 4, l. 23-24: very often, canopy transport is dominated by large eddies which violate gradient-diffusion theory

(6) p. 4, l. 29: I am confused by the sign convention here – does that mean that soil respiration has a negative sign?

(7) P. 4, l. 34: Eq. (2) being based on concentrations, has a huge footprint and thus integrates a much larger area compared to chamber measurements

(8) P. 5, l. 27: here an explanation/justification for the weighting with the light profile within the integral is in order

(9) P. 5, l. 30: Tl calculated this way is more commonly referred to as the aerodynamic temperature

(10) P. 6, l. 2, l. 9: how was the vertical averaging done?

(11) P. 6, l. 5: PAR is incident?

(12) P. 6, l. 24: wind direction-dependent

(13) P. 6, l. 33-35: this basically is the second method – should be described and placed more prominently

(14) P. 7, section 2.6: this is how you achieve closure in your system of equations – should be spelt out more prominently

(15) P. 8, l. 30-p. 9, l. 6: this is important and should be backed up with a graph showing COS and CO2 eddy and storage fluxes, not necessarily in the main text, but at least in the supplement

(16) P. 9, l. 20: big leaf – see De Pury & Farquhar (1997; PCE)

(17) P. 10, section 3.3: a strong point would be if the approach were able to pick up the differences between times when the canopy is wet after rain (larger evaporative fraction) and when the canopy is dry (mostly transpiration)

(18) P. 10, l. 23-30: this section would be easier to follow based on the suggested flow chart

(19) P. 10, l. 40: while you did not troubleshoot the models – but how did you determine the parameters?

———————————————————

---

## Referee Comment (RC2) · Anonymous Referee #2 · 2 Nov 2016

This is an excellent paper that I enjoyed reading, which shows novel application (!) of COS flux measurements to estimate canopy scale conductance. This is a welcome deviation from the current focus on using COS as tracer for CO2 uptake and GPP, to address another critical uncertainty in ecosystem research.

The paper is well written and provides a rigorous treatment of the topic. The results on the seasonal patterns in conductance and the ET partitioning, including the "counter-intuitive" trends is interesting and intriguing. I have a few comments that can be addressed in minor revisions before formal publication in BC, which is fully warranted.

[Figure]

– while the estimates of total canopy conductance with COS is straight forward, esti-
mating stomatal conductance is more tricky and is not necessarily well constrained by
COS alone. It seems the uncertainties in the modeling of the wide range of the addi-
tional parameters is somewhat played down and more indications of the uncertainties
involved should be made.

–In that sense, the statement that agreement between two methods validate both is a
bit strong. I think its "reassuring" (but we should remain cautious)...

–In particular, I found the discussion on the estimation of the biochemical conductance
interesting, but perhaps incomplete. There is little reference to the literature available
on the topic, both COS (e.g. Berry et al 2103, Stimler et al 2011, etc.), in much of the
CO2- 18O literature where similar aspects had to be addressed, and other physiologi-
cal studies. Ultimately, perhaps CA poses little resistance to COS uptake at all...

–Although the focus on conductance is commendable, the total ignorance of CO2 ex-
change (e.g. Fig. 1) is problematic and should clearly be addressed. This are important
constraints on COS estimates, and information on COS/CO2 relationships from such
study will be very valuable to other COS studies. In fact, it could also be interesting to
check if the COS-based g is valid for CO2 exchange, and perhaps to use results on
COS-based T, and CO2 to look at seasonal trends of WUE.

———————————————

---

## Author Comment (AC1) · 30 Nov 2016

Response of Authors Wehr et al. (hereafter W) to Anonymous Referee #1 (hereafter AR1):

AR1: This is one of, if not the, first study to use carbonyl sulfide (COS) to investigate canopy- scale stomatal conductance and transpiration (and by difference from evapotranspiration, evaporation). The authors are modelling the various resistances to canopy COS exchange, which they infer from ecosystem-scale COS flux measurements and estimates of the soil exchange, to finally back out the stomatal resistance/conductance. These estimates are compared against estimates derived from a combination of sensible and latent heat flux measurements. The two approaches agree reasonably well, except for around dawn, when the authors suspect measurement artefacts. Two, random, land surface models do a poor job in simulating the magnitude, partitioning and seasonal evolution of the water fluxes, highlighting the value of the presented data for improving models. This is an excellent paper, well written and innovative. The comments below serve to further improve the manuscript.

W: We thank the referee for taking the time to improve our manuscript, and for these supportive comments.

AR1: Major comments: (1) In contrast to the authors I believe that their approach does not rely "minimally on modelling" (p. 1, l. 24), but rather that they use a complex model to simulate tremendously complex processes. In particular the vertical heterogeneity of the microclimate and associated patterns in exchange processes constitute a real challenge for up- scaling. I found the description of the modelling approaches sometimes hard to follow, in particular with regarding to keeping track what the different inputs are and by which approach these are used and what the assumptions are. In order to make this easier for the reader to follow I am suggesting a sort of a flow chart or similar that shows the inputs for the two approaches, lists the main equations and outputs and so makes it easier to follow what the differences between the approaches are.

W: We agree that "minimally" was a poor choice of word; we meant that partitioning ET based on empirical, EC-derived stomatal conductance is much more direct and constrained than partitioning based on a full-blown ecosystem model. That is probably not a key point to make, and so we will simplify the sentence to read: "Our method of ET partitioning avoids concerns about mismatched scales or measurement types because both ET and transpiration are derived from eddy covariance data."

We like the idea of the flow chart; we have drafted one and attached it to this comment.

AR1: (2) I am surprised to see that what I would consider the standard approach for

assessing the surface conductance to water vapour, the Penman-Monteith combination equation, was not used. This approach is used a lot in the flux measurement community and it would be nice to see how it compares to the other approaches. In doing so, and the same applies to the approach based on sensible and latent heat fluxes, the authors will need to deal with the energy imbalance (if it exists at this site, which I though presume).

W: Energy imbalance does exist at this site, as at most eddy flux sites, which is one reason that we do not use the Penman-Monteith (PM) equation. The sensible and latent heat flux equations that we use do not require energy balance. The PM equation, on the other hand, is derived from the sensible and latent heat flux equations by using energy balance to eliminate the heat flux as a variable (and by making a couple of approximations). When sensible heat flux measurements are available, as at every eddy flux site, there is no reason to eliminate the heat flux from the equations or make approximations. Moreover, it is not reasonable to use energy balance to eliminate the heat flux when the available energy flux measurements (heat, water, radiation) do not satisfy energy balance.

We will add the following paragraph on this topic to Section 2.5:

"Note that Eqs. (6,9-10) are the basis for the Penman-Monteith (PM) equation (Monteith et al., 1965), which is commonly used to estimate stomatal conductance. The PM equation was derived from them by using energy balance to eliminate the heat flux as a variable, and by some approximations. When sensible heat flux measurements are available, as at every eddy flux site, there is no reason to eliminate the heat flux from the equations. Moreover, it is inadvisable to use energy balance to eliminate the heat flux given that at most eddy covariance sites, including ours, the available energy flux measurements (heat, water, radiation) do not satisfy energy balance (Wilson et al., 2002). This energy balance closure problem has been repeatedly investigated but the culprit remains unclear and might vary between sites (Wilson et al., 2002; Foken, 2006; Lindroth et al., 2010). In any case, eddy covariance determines the sensible heat flux

H at least as reliably as the water vapor flux E, and so there is no plausible scenario in which the PM equation would give a more accurate stomatal conductance than Eqs. (6,9-10). At our site, stomatal conductance obtained by inverting the PM equation was about 20% lower than that obtained from Eqs. (6,9-10)."

The cited references are:

Monteith, J. L., Szeicz, G., and Waggoner, P. E.: The Measurement and Control of Stomatal Resistance in the Field, J. Appl. Ecol., 2(2), 345-355, 1965.

Wilson, K., Goldstein, A., Falge, E., Aubinet, M., Baldocchi, D., Berbigier, P., Bernhofer, C., Ceulemans, R., Dolman, H., Field, C., Grelle, A., Ibrom, A., Law, B. E., Kowalski, A., Meyers, T., Moncrieff, J., Monson, R., Oechel, W., Tenhunen, J., Valentini, R., Verma, S.: Energy balance closure at FLUXNET sites, Agr. Forest Meteorol., 113, 223–243, 2002.

Foken, T.: The Energy Balance Closure Problem: An Overview, Ecol. Appl., 18(6), 1351–1367, 2008.

Lindroth, A., Mölder, M., and Lagergren, F.: Heat storage in forest biomass improves energy balance closure, Biogeosciences, 7, 301–313, 2010.

AR1: (3) In my view any paper should have a spelled-out statement of objectives and/or hypothesis and I am asking the authors to modify the introduction paragraph accordingly.

W: We will add the following sentence to the end of the first paragraph: "Our objective here is to test Wehr and Saleska (2015)'s method for estimating canopy stomatal conductance from the water vapor flux against a new, independent method based on carbonyl sulfide (OCS)—and then to use stomatal conductance to partition evapotranspiration."

AR1: Detailed comments: (1) p. 1, l. 32: and degree of opening

W: We will make this change.

AR1: (2) p. 2, l. 13: CO2 assimilation depends both on the light AND dark reactions

W: We were not neglecting the dark reactions; we were merely making the point that the rate of CO2 hydrolysis ultimately depends on light while the rate of OCS hydrolysis does not. To avoid confusion, we will alter the sentence to read: "...the net rate of CO2 hydrolysis depends on downstream reactions involving light while the rate of OCS hydrolysis does not."

AR1: (3) p. 2, l. 16: stomata are generally the most influential component

W: It seems the referee means to soften the statement to allow for the possibility that stomata are not always the most influential component. Rather than "generally", which can be understood to make the statement even stronger, we will add the word "typically".

AR1: (4) p. 3, l. 31: that will work if both COS and CO2 suffer the same attenuation; for CO2 and H2O we know that this does not work very well, H2O being more strongly attenuated in the inlet tubes

W: That is a good point. We will add the sentence: "This approach assumes that OCS and CO2 suffer the same attenuation, which would not be true for a 'sticky' molecule like H2O."

AR1: (5) p. 4, l. 23-24: very often, canopy transport is dominated by large eddies which violate gradient-diffusion theory

W: Agreed. To address this point, we will add the following sentences: "The assumption that the eddy transport near the ground could be treated as gradient-driven is questionable, as turbulent transport in forest canopies is dominated by large, quasi-periodic eddy motions (Raupach et al., 1996). Nonetheless, the consistency of the calculated soil OCS uptake over the diel cycle and over the growing season (Section 3.1) suggests that the assumption is sufficient for our purposes. Moreover, the calculated soil OCS

uptake varied by less than 0.2 pmol m-2 s-1 between low- and high-turbulence conditions as quantified by the friction velocity (results not shown)." The citation is: Raupach, M. R., Finnigan, J. J., and Brunet, Y.: Coherent eddies and turbulence in vegetation canopies: the mixing-layer analogy, Boundary-Layer Meteorology, 78, 351-382, 1996.

AR1: (6) p. 4, l. 29: I am confused by the sign convention here – does that mean that soil respiration has a negative sign?

W: Yes, that is the convention used, as stated after the equation: "where the F are fluxes into the ground (the $CO_2$ flux is negative)...".  The same convention (uptake fluxes are positive) is used throughout the paper.

AR1: (7) P. 4, l. 34: Eq. (2) being based on concentrations, has a huge footprint and thus integrates a much larger area compared to chamber measurements

W: We are not convinced of that. The effective footprint of near-ground concentration measurements (the lower inlet being just 20cm away from the soil) is unclear and probably depends on the wind conditions and density of understory foliage. To acknowledge this issue, we have extended the sentence, "The flux tower itself, where the gradients were measured, was located about 20 m to the north of the chambers", by adding the phrase, "and the effective footprint of the near-ground concentration measurements is unclear".

AR1: (8) P. 5, l. 27: here an explanation/justification for the weighting with the light profile within the integral is in order

W: For the reader's convenience, we will add the sentence: "The light absorption profile $\varphi(\zeta)$ is used here as a proxy for the heat source profile." Further details underlying the equations in this section are explained in the cited reference, Wehr and Saleska (2015).

AR1: (9) P. 5, l. 30: Tl calculated this way is more commonly referred to as the aerodynamic temperature

W: We will add the following sentence: "Surface temperatures estimated based on heat

flux in this way have been termed "aerodynamic temperatures" to contrast them with radiometric temperatures (Kustas et al., 2007)." The citation is: Kustas, W. P., Anderson, M. C., Norman, J. M., and Li, F.: Utility of radiometric–aerodynamic temperature relations for heat flux estimation, Boundary-Layer Meteorol., 122, 167–187, 2007.

AR1: (10) P. 6, l. 2, l. 9: how was the vertical averaging done?

W: We will clarify this point by adding the following: "The canopy airspace temperature $T_n$ was derived from the above-canopy air temperature using a turbulent eddy conductance derived in turn from the corresponding $CO_2$ gradient and NEE. The above-canopy $CO_2$ value was measured at 29 m and the within-canopy $CO_2$ value was the average of measurements at 12.7 m and 18.3 m (canopy top height is about 25 m). The 12.7 m and 18.3 m measurements were generally indistinguishable during the day due to efficient turbulent mixing."

AR1: (11) P. 6, l. 5: PAR is incident?

W: Yes; we will say "incident PAR" in our revised manuscript.

AR1: (12) P. 6, l. 24: wind direction-dependent

W: We will add the word "wind".

AR1: (13) P. 6, l. 33-35: this basically is the second method – should be described and placed more prominently

W: Yes, we should have been more clear about that. We will reorder this section and replace the sentence in question with the following: "Eqs. (8-9) represent the water-flux method for estimating stomatal conductance. To obtain stomatal conductance from our OCS measurements, we rearranged Eq. (1) to solve for $g_s$ and inserted the measured canopy OCS uptake F along with the boundary layer, mesophyll, and biochemical conductance estimates described in this section and in Section 2.6. As mentioned above (and see Section 3.2), the stomatal and biochemical conductances were limiting to OCS uptake at our site, and so the calculated stomatal conductance was not sensitive

to the boundary layer and mesophyll conductances used."

AR1: (14) P. 7, section 2.6: this is how you achieve closure in your system of equations – should be spelt out more prominently

W: We will begin Section 2.6 with this: "The one term in Eq. (1) that is not constrained by measurements, empirical models, or established theory is the biochemical conductance associated with carbonic anhydrase activity, gCA. Apparent CA activity depends on the amount of CA enzyme and on where it is located relative to the intercellular air spaces, but little is presently known about either of those things (Berry et al., 2013). We therefore tested two simple assumptions for gCA that allow us to solve Eq. (1)."

AR1: (15) P. 8, l. 30-p. 9, l. 6: this is important and should be backed up with a graph showing COS and CO2 eddy and storage fluxes, not necessarily in the main text, but at least in the supplement

W: We can show OCS and CO2 flux data that illustrates the dawn peak in NEE and in OCS uptake that is described in the text. A draft figure showing such data is attached to this comment and could be included in a supplement or in the main text, at the editor's discretion. In our opinion, showing the eddy and storage flux data separately would clutter the plot without helping the argument (since the problem we are talking about in any case is inaccurate measurement of the storage flux).

AR1: (16) P. 9, l. 20: big leaf – see De Pury & Farquhar (1997; PCE)

W: We will add the citation of De Pury and Farquhar.

AR1: (17) P. 10, section 3.3: a strong point would be if the approach were able to pick up the differences between times when the canopy is wet after rain (larger evaporative fraction) and when the canopy is dry (mostly transpiration)

W: The approach can indeed pick that up, and we will include a new figure showing the water fluxes versus time since rain. A draft figure is attached to this comment.

AR1: (18) P. 10, l. 23-30: this section would be easier to follow based on the suggested flow chart

W: We hope that the attached flow chart helps.

AR1: (19) P. 10, l. 40: while you did not troubleshoot the models – but how did you determine the parameters?

W: The ED2 and SiB3 parameters were optimized for the Harvard Forest using standard data that included NEE and ET, but not our partitioned water fluxes or OCS measurements. We will state this in our revised manuscript.

―――――――――――――――――

**Measurements**

OCS

Wind Speed

Air Temperature and Heat Flux

Water

Light

**Theoretical Model**

$g_b$

**Empirical Models**

$g_m$

$\boldsymbol{g_s}$

**Assumed**

Constant CA Activity

Leaf Temperature

**Derived Quantities**

Total OCS Conductance

Insensitive

Insensitive

Water Vapor Gradient

Fit During Dry Periods

$\boldsymbol{g_s}$

$g_s$

**Agreement**

**Fig. 1.** Flowchart

**Fig. 2.** Unmeasured storage example

[Figure]

**Fig. 3.** Water fluxes vs time since rain

---

## Author Comment (AC2) · 30 Nov 2016

Response of Authors Wehr et al. (hereafter W) to Anonymous Referee #2 (hereafter AR2):

AR2: This is an excellent paper that I enjoyed reading, which shows novel application (!) of COS flux measurements to estimate canopy scale conductance. This is a welcome deviation from the current focus on using COS as tracer for CO2 uptake and GPP, to address another critical uncertainty in ecosystem research. The paper is well written and provides a rigorous treatment of the topic. The results on the seasonal pat-

terns in conductance and the ET partitioning, including the "counter- intuitive" trends is interesting and intriguing. I have a few comments that can be ad- dressed in minor revisions before formal publication in BC, which is fully warranted.

W: We thank the referee for taking the time to improve our manuscript, and for these kind words.

AR2: – while the estimates of total canopy conductance with COS is straight forward, estimating stomatal conductance is more tricky and is not necessarily well constrained by COS alone. It seems the uncertainties in the modeling of the wide range of the additional parameters is somewhat played down and more indications of the uncertainties involved should be made. –In that sense, the statement that agreement between two methods validate both is a bit strong. I think its "reassuring" (but we should remain cautious)...

W: Indeed, there are sources of substantial systematic uncertainty in the OCS-flux method (mainly the CA activity) and in the water-flux method (mainly the assumptions about evaporation), and they are unfortunately difficult (perhaps even impossible) to rigorously quantify because they are mostly model structural uncertainties rather than measurement uncertainties. That is precisely the motivation for the article, for seeking validation. A key point of the paper is therefore that the two methods give the same answer for stomatal conductance even though very few sources of uncertainty are shared between them. In other words, our conclusions are not subject to the uncertainties in one method or the other, but rather to the uncertainties they have in common. That point is made on p.10, lines 23-30:

"Although our transpiration and evaporation estimates are not direct measurements, only three sets of assumptions are common to both of the methods by which we derived those estimates from our measurements: (1) the assumptions involved in the eddy covariance method, which seem to fail at dawn as discussed in Sect. 3.2; (2) the boundary layer resistance model described in Sect. 2.5, to which our estimates

are insensitive (within plausible bounds); and (3) the application of stoma-scale gas transport equations at the canopy scale, which neglects biological and environmental heterogeneity within the canopy but is a sufficiently good approximation to allow the observed temporal patterns of canopy OCS uptake to be precisely predicted from the observed heat and water vapor exchange, as shown in Sect. 3.1. Given the close agreement between the two otherwise independent methods (Sect. 3.2), our estimates of transpiration and evaporation should be accurate."

Hopefully the new flow chart attached to our response to Referee #1 will help readers to see what information goes into each method.

Regarding in particular "modeling of the wide range of the additional parameters", there are 3 modeled parameters involved in going between total canopy OCS conductance (i.e. the OCS measurements) and stomatal conductance: (1) the boundary layer conductance, (2) the mesophyll conductance, and (3) the biochemical conductance. The first two are estimated to be about 3X larger than the peak stomatal conductance, which is therefore not sensitive to them (a point that we will emphasize in our revised manuscript). For example, based on the conductances shown in Fig. 3, the total resistance at midday is about 40% gs, 40% gCA, 10% gm and 10% gb; so a 10% bias in gm or gb would cause only a 2.5% bias in gs. The biochemical conductance (i.e. CA activity) is the overwhelming source of uncertainty, since it is simply assumed–but again, confidence is obtained by agreement between the diel and seasonal patterns obtained from the two largely independent methods.

AR2: –In particular, I found the discussion on the estimation of the biochemical conductance interesting, but perhaps incomplete. There is little reference to the literature available on the topic, both COS (e.g. Berry et al 2103, Stimler et al 2011, etc.), in much of the CO2- 18O literature where similar aspects had to be addressed, and other physiological studies. Ultimately, perhaps CA poses little resistance to COS uptake at all...

W: Thank you for pointing out the lack of supporting literature for our discussion of CA activity, which was an oversight on our part. The salient point for the present manuscript seems to be that the literature on the topic does not give us any means to estimate CA activity. We will therefore change the methods section on CA activity to begin as follows: "The one term in Eq. (1) that is not constrained by measurements, empirical models, or established theory is the biochemical conductance associated with carbonic anhydrase activity, gCA. Apparent CA activity depends on the amount of CA enzyme and on where it is located relative to the intercellular air spaces, but little is presently known about either of those things (Berry et al., 2013)."

AR2: –Although the focus on conductance is commendable, the total ignorance of CO2 exchange (e.g. Fig. 1) is problematic and should clearly be addressed. This are important constraints on COS estimates, and information on COS/CO2 relationships from such study will be very valuable to other COS studies. In fact, it could also be interesting to check if the COS-based g is valid for CO2 exchange, and perhaps to use results on COS-based T, and CO2 to look at seasonal trends of WUE.

W: As the referee is no doubt aware, the total OCS conductance g in Eq. (1) is not strictly valid for CO2 because CO2 uptake is not only a matter of diffusion and CA activity, but also of Rubisco and especially light, as discussed on page 2, lines 12-13. In other words, the leaf relative uptake (LRU, commonly discussed in the OCS literature) is not constant. However, it may be approximately constant–perhaps enough for certain applications–and there is of course much interest in using OCS as a proxy for GPP. So, although this article is about stomatal conductance and ET (not GPP or even OCS per se), we will insert this new, short section on the GPP-OCS relationship for interested readers:

"3.3 Gross Primary Productivity and Leaf Relative Uptake

The success of Eq. (1) in predicting canopy OCS uptake supports the theoretical expectation that canopy OCS uptake is dependent on the atmospheric OCS mixing

ratio, the diffusive conductance, and CA activity–and is therefore not directly related to GPP, which depends also on the enzyme Rubisco and on light. There is nonetheless widespread interest in using OCS uptake as a proxy for GPP. So for interested readers, we show how GPP and OCS uptake compare over the diel cycle and over the growing season in Figs. NEW and NEW, using both isotopic and standard partitioning methods for estimating GPP as described in Wehr et al. (2016). We include a common metric for comparison, the leaf relative uptake (LRU): LRU is defined as the ratio of the canopy OCS uptake to GPP, divided by the ratio of the atmospheric OCS concentration to the atmospheric CO2 concentration. Figs. NEW and NEW show that, as expected, LRU is inversely correlated with PAR, as PAR can directly limit uptake of CO2 but not of OCS."

Drafts of the two new figures mentioned are attached to this comment.

Wading into water-use efficiency, while interesting, would be considerably off-topic, and we would prefer not to do so. Intrinsic WUE using the same water-flux-based estimates of stomatal conductance as here has already been reported in Wehr et al., Seasonality of Temperate Forest Photosynthesis and Daytime Respiration, Nature 534, 680-683, doi:10.1038/nature17966, 2016.

[Figure]

**Fig. 1.** Diel cycle of GPP and leaf relative uptake

[Figure]

**Fig. 2.** Seasonal cycle of GPP and leaf relative uptake

---

## Author Response (AR1)

Dear Georg,

We are pleased to submit our revised manuscript, following your and the referees' suggestions, along with a marked up copy in which all changes since the original submission are detailed in red. Below you will also find our response to your comment. We have not included a separate list of all changes because it would be redundant with (and surely less useful than) the marked up manuscript.

We thank you for your efforts on our manuscript. Please let us know if you would like anything else from us.

Best Regards,

Rick (on behalf of the coauthors)

**Response to the Associate Editor of bg-2016-365**

**Response of Authors Wehr et al. (hereafter W) to the Associate Editor (hereafter AE):**

*AE: Both reviews were very much favorable of ms bg-2016-365 and the author reply is reasonable (with maybe one exception detailed below) and the new figures will contribute towards improving an already excellent manuscript. I am thus inviting the authors to submit a revised version.*

*The one issue where the authors did not entirely convince me regards the comment on energy balance closure raised by reviewer #1. True that the approach the authors have taken does not assume energy balance closure, but as the energy balance is not closed at their site, this indicates that at least one of the energy balance terms is underestimated (H, LE, G) or overestimated (Rnet). Following the author's argumentation they implicitly assume that H and LE they use are measured correctly and as a consequence the energy imbalance at this site is due to errors in Rn and G (and other non-turbulent terms)*

W: Thank you very much for helping to improve our manuscript. We have submitted a revised draft for your consideration, based on our replies to the referee comments.

Evidently, we were not clear enough in our response to Referee #1 about energy balance. We do not assume that energy imbalance is due to errors in the non-turbulent terms when advising against the PM equation. Rather, our point is that the PM equation is less accurate than our equations regardless of which terms in the energy balance are misestimated. This revised ending to the relevant paragraph is more detailed and will hopefully clarify:

[revised manuscript text omitted]

Richard Wehr 11/18/2016 4:26 PM

Richard Wehr 11/18/2016 4:29 PM

Richard Wehr 11/18/2016 4:29 PM

Richard Wehr 11/18/2016 4:27 PM

Richard Wehr 11/18/2016 4:27 PM

Richard Wehr 11/18/2016 4:27 PM

Richard Wehr 11/18/2016 4:29 PM

Richard Wehr 11/18/2016 4:10 PM

Richard Wehr 11/18/2016 4:29 PM
**Moved up [1]:** The mesophyll resistance to $CO_2$ transport was calculated as:

$$r_m^{CO_2} = \frac{P}{RT_L}\left(\text{LAI}\times 0.188 \times \exp\left(-\frac{1}{2}\left(\frac{\ln((T_L-273.15)/28.8)}{0.610}\right)^2\right)\right)$$
.........(10)

following the empirical temperature response for *Quercus canariensis* measured by Warren and Dreyer (Warren and Dreyer, 2006), which agrees with the value determined by Manter and Kerrigan (Manter and Kerrigan, 2004) at 25°C for *Quercus rubra* (red oak), which dominates our site (Wehr and Saleska, 2015). There is debate in the literature as to whether mesophyll conductance varies also in response to light, humidity, and other rapidly changing environmental variables (Flexas et al., 2012; 2008; Tholen et al., 2012). Some studies have argued that mesophyll conductance co-varies with stomatal conductance on a timescale of minutes (Flexas et al., 2008), whereas others have argued that an apparent correlation between the two conductances results from failing to account for diffusion of $CO_2$ from the mitochondria to the chloroplasts (Tholen et al., 2012). Because the stomatal and biochemical conductances limit OCS uptake at our site, changing the mesophyll conductance to be perfectly correlated with the stomatal conductance (but with a mean value given by Eq. (10)) has a negligible effect on the predicted OCS uptake from Eq. (1). We thus were unable to use OCS fluxes to directly address debates about controls on mesophyll conductance.

Richard Wehr 11/21/2016 9:56 AM

Richard Wehr 11/18/2016 4:33 PM

Richard Wehr 11/18/2016 4:33 PM

[revised manuscript text omitted]

Richard Wehr 11/19/2016 12:27 PM
**Comment [4]:** New figure

---

## Author Response (AR2)

Dear Georg,

We hope that our new revision and our response to your comments below will better address the desire for a comparison to the standard Penman-Monteith approach to estimating stomatal conductance.

Best Regards,

Rick (on behalf of the coauthors)

**Response to the Associate Editor of bg-2016-365**

Response of Authors Wehr et al. (hereafter W) to the Associate Editor (hereafter AE):

AE: The revised manuscript nicely incorporates the changes recommended by the two reviewers and I am thus conditionally accepting it.

The issue that the authors in my view still do not get quite right is again the reasoning why they didn't adopt the Penman-Monteith (PM) combination equation (l. 21-36 on p. 7 or revised ms).

While the PM model was originally formulated in terms of LE (elimitating H), being based on the principle of energy balance closure, it can be expressed in many different ways. For example, the PM model can be written in terms of the Bowen-ration (see Wohlfahrt et al. 2009, 10.1016/j.agrformet.2009.03.015). The major difference, in my view, between the PM model and the approach the authors have taken is that the former makes an implicit assumption about energy balance closure. The original version assigns any missing energy implicitly to H (as the authors correctly point out), while the version based on the Bowen ration distributes any missing energy to both H and LE according to the Bowen ratio (which satisfies the author's reasoning that both H and LE are likely to be affected by the energy imbalance). In the latter case, LE and thus surface conductance will be larger.

So in summary I do not see why the PM model would systematically yield lower values for gs. One good reason for not choosing the PM model would be the desire not to make assumptions about energy balance closure, also because then one probably would need to think about adjusting the COS flux for a possible underestimation (if the cause for the energy imbalance causes an underestimation of the COS flux as well).

W: We were taking "the Penman-Monteith equation" to mean the equation derived by Penman and Monteith (i.e. Monteith, 1965). That equation, our equations, and Eq. (2) of Wohlfahrt et al. (2009) are all variations on the same model, differing only in how the sensible and latent heat fluxes are estimated. Hopefully the following revision to the paragraph in question will resolve the issue without too much of a digression into the literature on energy balance closure:

"Eqs. (6.9-10) can be combined to obtain the Penman-Monteith (PM) equation (Monteith et al., 1965), which is commonly used to calculate stomatal conductance. In principle, the two approaches are equivalent. However, the PM equation expresses *H* as the residual of the other terms in the energy budget (i.e. net radiation  $R_{n}$ , latent heat flux  $\lambda E$ , conductance to ground G, and energy storage S) because measurements of *H* were not available when the PM equation was derived. As *H* is measured at all eddy flux sites, using the PM equation to calculate stomatal conductance from eddy flux measurements of *E* is unnecessarily circuitous. Moreover, it yields a different result in practice because the measured energy budget is incomplete at most eddy flux sites, including ours, with the radiative input exceeding the turbulent output by about 20%. This energy balance closure problem has been repeatedly investigated but the culprit remains unclear and might vary between sites (Wilson et al., 2002; Foken, 2008; Lindroth et al., 2010). If the closure problem were known to be due to low bias in the eddy covariance method, as suggested by Foken (2008), then it would be appropriate to adjust H and E to satisfy energy balance while maintaining the measured Bowen ratio ( $H/\lambda E$ ), either by modifying the PM equation as in Wohlfahrt et al. (2009) or by multiplying H and E by  $(R_n - G - S)/(H + \lambda E)$  in Eqs. (6,9); however, recent evidence suggests that the problem is due to energy storage in the soil and biomass instead (Lindroth et al., 2010). At our site, the original PM equation underestimates stomatal conductance by 20% compared to Eqs. (6,9-10)."

**Dynamics of canopy stomatal conductance, transpiration, and evaporation in a temperate deciduous forest, validated by carbonyl sulfide uptake**

Richard Wehr1, Róisín Commane2, J. William Munger2, J. Barry McManus3, David D. Nelson3, Mark S. Zahniser3, Scott R. Saleska1, Steven C. Wofsy2

[revised manuscript text omitted]

Richard Wehr 11/18/2016 4:31 PM Deleted: 10

Richard Wehr 11/18/2016 4:29 PM

Richard Wehr 11/18/2016 4:29 PM

Richard Wehr 11/18/2016 4:29 PM

Moved (insertion) [1]

(8)

(6)

(7)

where PAR is incident photosynthetically active radiation ( $\mu$ E m-2 s-1), VPG is the water vapor pressure gradient (Pa) between the canopy airspace and the substomatal cavity, and  $\chi$  is the clear sky index (i.e. the ratio of observed PAR to the PAR that would have been observed for the same solar zenith angle under clear skies). The coefficients  $b_0$ ,  $b_1$ , and  $b_2$  are empirically determined by fitting Eq. (a) to the following water vapor flux-gradient equation during times when water vapor flux was dominated by transpiration (i.e. evaporation was minimal, typically several days after rain, excluding mornings with possible

$$r_s^{\rm H_2O} = \frac{\rm VPG}{\rm RT_{nE}} - r_b^{\rm H_2O} , \label{eq:rs}$$

dew evaporation):

[revised manuscript text omitted]

Richard Wehr 11/18/2016 4:26 PM Deleted: is Richard We hr 11/18/2016 4:29 PM Deleted: 7 Richard We hr 11/18/2016 4:29 PM Deleted: 8 hard Wehr 11/18/2016 4:27 PM Deleted: cannot Richard Wehr 11/18/2016 4:27 PM Deleted: s Richard Wehr 11/18/2016 4:27 PM Deleted: is Richard Wehr 11/18/2016 4:29 PM Deleted: Richard Wehr 11/18/2016 4:10 PM Deleted: We also calculated daytime stomatal conductance by rearranging Eq. (1) and inserting the measured canopy OCS uptake F and the other conductance estimates described in this section Richard Wehr 11/18/2016 4:29 PM Moved up [1]: The mesophyll resistance to CO2

**transport was calculated as: $r \frac{CO}{m^2} = \frac{P}{RT L} (LAI \times 0.188 \times 10^{-2})$**

 $\frac{1}{RT} \int_{L} \left( LAI \times 0.188 \times \exp\left(-\frac{1}{2} \left(\frac{\ln((T L - 273.15))/28.8}{0.610}\right)^{-2}\right)^{-2}\right)^{-2} \right)^{-2}$

following the empirical temperature response for Quercus canariensis measured by Warren and Dreyer (Warren and Dreyer, 2006), which agrees with the value determined by Manter and Kerrigan (Manter and Kerrigan, 2004) at 25°C for Quercus rubra (red oak), which dominates our site (Wehr and Saleska, 2015). There is debate in the literature as to whether mesophyll conductance varies also in response to light, humidity, and other rapidly changing environmental variables (Flexas et al. 2012; 2008; Tholen et al., 2012). Some studies have argued that mesophyll conductance co-varies with stomatal conductance on a timescale of minutes (Flexas et al., 2008), whereas others have argued that an apparent correlation between the two conductances results from failing to account for diffusion of CO2 from the mitochondria to the chloroplasts (Tholen et al., 2012). Because the stomatal and biochemical conductances limit OCS uptake at our site, changing the mesophyll conductance to be perfectly correlated with the stomatal conductance (but with a mean value given by Eq. (10)) has a negligible effect on the predicted OCS uptake from Eq. (1). We thus were unable to use OCS fluxes to directly address debates about controls on mesophyll conductance.

Richard Wehr 11/21/2016 9:56 AM Formatted: English (US)

Richard Wehr 11/18/2016 4:33 PM Deleted: 8 Richard Wehr 11/18/2016 4:33 PM Deleted: 7

covariance data, minimizing concerns about mismatched scales or measurement types. Such concerns arise when using sap flux sensors or gas exchange chambers to estimate transpiration, for example (Kool et al., 2014).

**3 Results and Discussion**

**3.1 Soil-atmosphere exchange of OCS**

- 5 The soil was a consistent sink of OCS at this site (Figs. 2 and 3). OCS uptake by soil was highly stable, with a mean of 2.2 pmol m-2 s-1 and no discernable diel or seasonal patterns (for May through October). That value is roughly in the middle of the range reported for temperate forest soils (-8 to +1.45 pmol m-2 s-1) (Whelan et al., 2015). Laboratory incubations of soy field soil have shown OCS uptake to depend strongly on soil temperature and water content (Whelan et al., 2015), but we see no indication of those dependencies in the field at our site. It may be that the dependencies are difficult to detect because the ranges of soil
- 10 temperature (~9-18°C) and volumetric water content (~22-37%) are too small. It may also be that the dependencies on temperature and water offset one another: OCS *production* increased with temperature in the incubations (Whelan et al., 2015) and might therefore be expected to be highest midseason, while OCS *consumption* peaked at roughly 20% water content in the incubations (Whelan et al., 2015) and might therefore also be expected to be highest midseason at our site.

**3.2 Canopy stomatal conductance and OCS uptake**

- 15 Nighttime uptake of OCS by the canopy averaged 2.4 pmol m-2 s-1, consistent with a nighttime stomatal conductance to OCS transport of 0.0016 mol m-2 s-1 on a leaf area basis (about 1/40 of the mean midday value). We used this value to generate the nighttime points on the predicted uptake curve in Fig. 2. The corresponding conductance to water vapor transport is 0.0031 mol m-2 s-1. Nighttime stomatal conductance to water vapor transport in individual red oak and red maple leaves at this site was previously reported to be no more than 0.05 mol m-2 s-1 (Daley and Phillips, 2006), in agreement with our result. At another site, measured values ranged from 0 to 0.039 mol m-2 s-1 with an average of 0.011 mol m-2 s-1 (Barbour et al., 2005), which is about 4
- times larger than found here using OCS. Previous estimates of nighttime *ecosystem* OCS uptake at this site averaged about 4 pmol m-2 s-1 between April and November (Commane et al., 2015), similar to the 4.6 pmol m-2 s-1 (= 2.2 soil + 2.4 canopy) found here.
- 25 During the day, Eq. (1) successfully predicted the diurnal and seasonal patterns of canopy OCS uptake when CA activity was assumed constant at 0.055 mol m-2 s-1 on a leaf area basis (Figs. 2 and 3, dotted lines), except at dawn (see below). Apart from dawn, the agreement with observations was within  $\pm 3\%$  for the diurnal pattern and  $\pm 10\%$  for the seasonal pattern (1 standard deviation). Assuming CA activity to be a function of temperature did not improve the agreement (Figs. 2 and 3, dashed lines). Rearranging Eq. (1) to solve for  $g_s$  using the measured canopy OCS uptake gave a diurnal cycle of stomatal conductance that
- 30 agreed with that obtained from Eq. (8) to within ±6% (1 standard deviation), again apart from dawn (Fig. 4).

At dawn, no value of CA activity or mesophyll conductance or boundary layer conductance was capable of bringing the predicted canopy uptake into agreement with the observations, because the stomatal conductance was strongly limiting at that time of day (Fig. 4). To obtain agreement, the stomatal conductance would have had to be 5 times larger than estimated from the measured heat and water fluxes, which is implausible. We therefore suspect that the apparent surge in canopy OCS uptake at dawn is a measurement artifact resulting from a failure of the vertical profile measurements on the flux tower to adequately capture changes in OCS storage within the canopy and subcanopy airspaces in the eddy covariance footprint (which is tens to

9

Richard Wehr 11/18/2016 4:42 PM Deleted: 1 Richard Wehr 11/18/2016 4:42 PM Deleted: 2

Richard Wehr 11/18/2016 4:42 PM Deleted: 1

| Richard Wehr 11/18/2016 4:42 PM |
|---------------------------------|
| Deleted: 1                      |
| Richard Wehr 11/18/2016 4:42 PM |
| Deleted: 2                      |
| Richard Wehr 11/18/2016 4:42 PM |
| Deleted: 1                      |
| Richard Wehr 11/18/2016 4:42 PM |
| Deleted: 2                      |
| Richard Wehr 11/18/2016 4:33 PM |
| Deleted: 7                      |
| Richard Wehr 11/18/2016 4:42 PM |
| Deleted: 3                      |
| Richard Wehr 11/18/2016 4:43 PM |
| Deleted: 3                      |

hundreds of meters upwind of the tower). Such a failure could occur because of differences in storage between the eddy covariance footprint and the profile footprint, and/or because of hidden OCS storage in unsampled airspaces, such as in the leaf litter layer or within tree crowns. Specifically, we speculate that our vertical profile measurements on the tower do not adequately capture the depletion of the OCS concentration in the canopy and subcanopy airspaces overnight and—more

- 5 importantly here—the replenishing of that concentration with increased turbulence at dawn. Thus eddy flux that is in reality serving to replenish the OCS concentration is misinterpreted as being taken up by the leaves. We have sometimes observed a similar phenomenon in CO2, but with reversed sign: CO2 accumulates in the canopy and subcanopy airspaces overnight and is flushed out at dawn, so that if the changes in canopy and subcanopy CO2 concentration are not adequately measured, there appears to be a dawn pulse in respiration and in the net ecosystem-atmosphere CO2 exchange.
- 10

40

The total conductance g, which controls the OCS uptake, is limited by  $g_s$  early and late in the day but co-limited by  $g_s$  and  $g_{CA}$  through the middle of the day, so that g has a much flatter diurnal peak than  $g_s$  (Fig. 4; note the log scale). Eq. (1) must therefore include the correct diurnal patterns and magnitudes of both stomatal conductance and CA activity in order to reproduce the observed diurnal pattern of canopy OCS uptake. Notwithstanding the dawn artifact, then, the close agreement between the

temporal patterns of observed and predicted stomatal conductance in Fig. 4 (or total conductance in Fig. 4 or OCS uptake in Fig. 2) constitutes a strong validation of the simple empirical description of canopy stomatal conductance in Eq. (8), which should provide a reliable means to partition ET at this mesic forest site.

Eq. (a) was distilled from a more general equation allowing for direct temperature and soil moisture effects on gs, and for a saturating response of gs to PAR, but those effects were found to be negligible (Wehr and Saleska, 2015). The linear response of (canopy-scale) gs to PAR in Eq. (b) contrasts with the saturating response generally observed at the leaf scale, and implies that leaf-level stomatal models like the Ball-Berry model (Collatz et al., 1991) or models based on optimality theory (Medlyn et al., 2011) should not be applied directly at the canopy scale (as in a 'big leaf' canopy model; De Pury and Farquhar, 1997), at least at this site. Note that even though canopy gs responds linearly to PAR, it will appear to show a saturating response when the vapor pressure gradient is not controlled for, as PAR and the vapor gradient tend to be positively correlated (because clouds and

- moisture go together) while  $g_s$  and the vapor gradient are negatively correlated (because stomata close to limit plant water loss). The insensitivity of  $g_s$  to soil moisture is somewhat surprising, but as Harvard Forest is a mesic site, it might be that soil water does not become limiting to the large trees even in a dry year such as 2012. The insensitivity to temperature, on the other hand, is not surprising: it has been argued (Lloyd and Farquhar, 2008) that photosynthesis declines at high temperature almost entirely
- 30 due to the indirect effect of temperature on stomatal conductance via the leaf-to-air water vapor pressure gradient, and Eq. (2) is consistent with that view.

The boundary layer conductance  $g_b$  is likely overestimated at night in Fig. 4 because the value from Eq. (5) is weighted vertically according to the profile of light absorption calculated from the leaf area density profile and the solar zenith angle—a method appropriate to sunlight-driven heat and CO2 fluxes during the day but not to any flux at night. Because the stomatal conductance is negligible at night and therefore strongly limiting, this overestimation of  $g_b$  is inconsequential here.

While treating CA activity as constant on a leaf area basis was sufficient to reproduce the observed canopy OCS uptake at this site, seasonal variations in CA activity at the  $\pm 15\%$  level could also be consistent with our observations, and might be important for some studies.

10

**Richard Wehr 11/18/2016 4:43 PM Deleted: 3**

| Richard Wehr 11/18/2016 4:43 PM |
|---------------------------------|
| Deleted: 3                      |
| Richard Wehr 11/18/2016 4:43 PM |
| Deleted: 3                      |
| Richard Wehr 11/18/2016 4:43 PM |
| Deleted: 1                      |
| Richard Wehr 11/18/2016 4:33 PM |
| Deleted: 7                      |
| Richard Wehr 11/18/2016 4:34 PM |
| Deleted: 7                      |
| Richard Wehr 11/18/2016 4:34 PM |
| Deleted: 7                      |

Richard Wehr 11/18/2016 4:34 PM Deleted: 7

[revised manuscript text omitted]

Richard Wehr 11/18/2016 4:44 PM Deleted: 2